# Regulation of *Vicia faba* L. Response and Its Effect on *Megoura crassicauda* Reproduction under Zinc Stress

**DOI:** 10.3390/ijms24119659

**Published:** 2023-06-02

**Authors:** Si-Jing Wan, Hui-Ru Si, Xian-Zhong Wang, Lei Chao, Wu Ma, Si-Si Sun, Bin Tang, Xiao-Ling Tan, Shigui Wang

**Affiliations:** 1College of Life and Environmental Sciences, Hangzhou Normal University, Hangzhou 311121, China; wsjw9898@163.com (S.-J.W.); 19550209390@163.com (H.-R.S.); wangxz9264@163.com (X.-Z.W.); chaolei0926@163.com (L.C.); 2021210315148@stu.hznu.edu.cn (W.M.); tbzm611@hznu.edu.cn (B.T.); 2Guizhou Institute of Mountainous Environment and Climate, Guiyang 550002, China; sunsisi3s@foxmail.com; 3State Key Laboratory for Biology of Plant Diseases and Insect Pests, Institute of Plant Protection, Chinese Academy of Agricultural Sciences, Beijing 100193, China

**Keywords:** *Vicia faba* L., *Megoura crassicauda*, trehalose Zn stress, bioaccumulation, heavy metal pollution

## Abstract

The heavy metal zinc (Zn) is known to be transmitted in the food chain; however, the effect of Zn stress on beans and herbivorous insects is largely unclear. This study aimed to investigate the resistance of broad bean plants to Zn stress and the consequent changes in their physiological and biochemical metabolism by simulating heavy metal pollution in soil. Simultaneously, the effects of aphid progeny treated with different Zn concentrations on the expression of carbohydrate and related genes were analyzed. The results showed that Zn had no effect on the germination rate of broad beans, but other effects mainly manifested as follows. (1) Chlorophyll content decreased. (2) The total soluble sugar and Zn content in stems and leaves increased with increasing Zn content. (3) The proline content first increased and then decreased with increasing Zn content. (4) The height of the seedlings indicates that low concentrations promote growth and high concentrations inhibit growth. In addition, only the first-generation fecundity decreased significantly when aphids fed on heavy metal broad beans. Continuous high Zn levels increase the trehalose content of aphid F1 and F2, while F3 decreases. These results can not only provide a theoretical basis for exploring the impact of soil heavy metal pollution on ecosystems but also preliminarily evaluate the possibility of broad beans as a means of pollution remediation.

## 1. Introduction

The continuous acceleration of China’s industrialization and uncontrolled anthropogenic activities have intensified and broadened the scope of heavy metal pollution of soil, which has exceeded the self-purification capacity of the ecological environment [1]. Surveys and research in China have shown that the main heavy metals are lead (Pb), cadmium (Cd), mercury (Hg), and arsenic (As) [2,3], which are generally characterized by latency, hysteresis, strong toxicity, non-degradability, and irreversibility. Metal enrichment in soil reduces soil fertility and quality, damages the ecological structure of the soil, and disrupts the balance of biological communities finally [4,5]. Zn, an essential element for plant growth and development, participates in the synthesis of various plant enzymes. However, excess Zn can cause irreversible damage to plant roots and inhibit photosynthesis and growth [6,7].

Amino acids are essential for plant growth. For example, the content of various amino acids can increase or decrease under certain stress conditions or food and host changes [8,9,10,11]. Additionally, alanine and other precursor amino acids of gluconeogenesis, which increase in the content in plants, indicate that protein decomposition and the gluconeogenesis pathway activity increase. This leads to increased sugar content, which provides energy for plants that rely on glucose for energy [12,13]. Some soluble sugars, such as glucose, sucrose, and galactose, increase in content at the initial stage of heavy metal stress and play an important protective role; subsequently, the content decreases with the continuation or enhancement of stress [14]. The glycolytic cycle of a plant in response to different heavy metal stresses is different and is related to the concentration and type of heavy metal stress [15]. Similarly, glucose is the main functional substance in insects, but trehalose is the main participant in the hemolymph. Trehalose, an insect blood sugar, is a non-reducing disaccharide. Trehalose is not only an energy storage material that provides energy for life activities but also an important protective factor that helps organisms to resist adversities such as dryness, high humidity, and low temperature and oxidation [16,17,18,19,20]. In insects, trehalose is mainly synthesized by trehalose-6-phosphate synthase (TPS) and trehalose-6-phosphate phosphatase (TPP), and it degraded into two molecules of glucose by the action of the trehalase (TRE) [21,22]. Trehalose is then transported to the hemolymph through specific transmembrane transport proteins, thereby maintaining the balance of carbohydrate metabolism in the body [23].

Legume crops have a strong ability to accumulate heavy metals and alleviate heavy metal pollution, such as Cd, in soil [24,25]. As the world’s third-largest spring food crop, broad beans can be used to repair heavy metals in the soil together with rhizobia [25,26,27]. Compared with other crops such as wheat, broad bean is also known for its rapid growth, high biomass with rich genotypes, and ability to accumulate and translocate Cd and Pb in various tissues. Many studies have also shown that leguminous plants adapt to the growth of heavy metal environments [28]. For example, low Cd pollution does not significantly inhibit the growth of broad bean seedlings [29]; Zhang’s experimental results indicate that Pb^2+^ has no significant effect on the germination of broad bean seeds [30]. The successful growth and significant yield of broad beans in copper (Cu), Cd, and Pb co-polluted soil indicate that broad beans have sufficient tolerance to mild to moderate Cd and Pb pollution, which indicated that leguminous plants can serve as excellent plants for repairing heavy metal polluted environments. During their growth, broad beans are often susceptible to pests, especially aphids [31], which are a common source of food for various predators and parasites [32,33,34,35,36]. Aphids are common pests with small size, strong fecundity, and parthenogenesis [31,37,38]. Their feeding is mainly focused on the inner side of plant leaves and fruit ears, and they absorb juice through the sieve tube of the phloem. Heavy metals can accumulate in the juice and are transferred to the aphids via feeding [39]. Heavy metals have toxic effects on insects [40]; in severe cases, they can cause cell apoptosis and have adverse effects on the growth index, development duration, mortality rate, and appetite of larvae and adults [41,42]. Research has shown that plant carbohydrates play an important role in the plants and insects [43].

A large number of studies have revealed the effects of heavy metal stress on plants, but they generally only focus on the physiological, biochemical, and stress resistance of plants. In addition, most of the heavy metal stresses and Hoagland nutrient solution cultivation started from the seedlings [44]. With the spread of the food chain, there is still limited research on heavy metals affecting their herbivorous insects. However, we conducted stress and simulated soil heavy metal pollution from the beginning of broad bean seeds. Hence, this study focused on the resistance of broad bean plants to heavy metal stress and whether heavy metal stress alters their physiological and biochemical metabolism, and we also investigated changes in the fecundity of aphids caused by the heavy metal pollution of broad beans. Some of the effects of excessive Zn stress on broad beans were explored; the effect of Zn stress on the trehalose metabolism of aphids in *Vicia faba* L. plants was also investigated. When continuously exposed to Zn, the examination of the physiological and biochemical effects on broad beans and the reproductive capacity of aphids can not only provide help for the preliminary assessment of phytoremediation of heavy metal pollution but also lay a foundation for further exploring the molecular mechanism of Zn homeostasis in plants and insects under Zn stress at the cellular and subcellular levels. In addition, it can provide a theoretical basis and experimental reference for further exploring the molecular mechanisms of heavy metal transfer and accumulation in the food chain.

## 2. Results

This experiment simulated heavy metal pollution by setting three different concentrations of Zn50, 100, and 150 mg/L, and it measured the germination rate, soluble sugar, proline, chlorophyll content, seedling height, and Zn accumulation of broad beans, which aim to evaluate broad bean’s resistance to Zn stress. In addition, the transmission of Zn through the food chain and the impact of Zn accumulation on the trehalose metabolism mechanism of the *M. crassicauda* were also studied.

### 2.1. Zn Concentration in Field Collection Sites and Broad Bean Roots, Stems, and Leaves after Zn^2+^ Irrigation

Zn content ranged from approximately 50 to 100 mg/L in the soil; therefore, we selected three concentrations of Zn (50, 100, and 150 mg/L) for our experiment. The Zn content in the broad bean stems was significantly different from each other, in the order of construction sites > campuses > vegetable fields, which may be due to construction phenomena (Figure 1A).

The Zn content in the roots and stems of broad beans increased with increasing Zn^2+^ irrigation concentration (Figure 1C,D). However, within a certain range, the Zn content in the roots increased with increasing Zn^2+^ irrigation concentration. When the Zn concentration was 150 mg/L, the Zn content in the roots of broad beans decreased (Figure 1B).

### 2.2. Effects of Zn Stress on Germination, Seedling Height, and Chlorophyll Content of Broad Bean

On days 9, 11, 13, and 15 of planting the broad beans, the seedling height in the 50 mg/L Zn treatment group was significantly higher than that of the control group. In addition, on day 9, the seedling height in the 150 mg/L Zn group was significantly higher than that of the control. On day 11, the seedling height in the 150 mg/L Zn group was significantly lower than that in the control group (Table 1). On days 13 and 15, seedling height in the 100 mg/L Zn group was significantly lower than that in the control group. Overall, the seedling height of the broad beans showed that low concentrations of Zn increased the height, whereas high concentrations of Zn decreased the height. From the broad bean phenotype, it was evident that the root systems in the heavy metal groups were significantly more developed than that control group at the first two time intervals. There were no significant differences over time (Figure 2A).

The germination rate of broad beans in each group showed a trend of initially being fast, then slowing with time, on days 7–9. However, overall, there was no significant difference in the germination rate between the Zn and control groups (Figure 2B). The chlorophyll a, b, and a + b contents in the broad bean leaves were lower than those in the control group, and there was a significant difference in the 50 mg/L Zn group (Figure 2C).

### 2.3. Effect of Zn Stress on the Content of Soluble Total Sugar and Proline (PRO) in Broad Beans

In the roots of broad beans, the soluble total sugar content of the different treatment groups was lower than that of the control group, and there was a significant difference between the 50 and 100 mg/L Zn groups. There was no significant difference in the total soluble sugar content of broad bean stems between the groups. In broad bean leaves, the total soluble sugar content of the different heavy metal treatment groups was higher than that of the control group, and there were significant changes in the content of the 100 and 150 mg/L Zn groups (Figure 3A). The PRO content in the roots, stems, and leaves of broad beans treated with different Zn concentrations was significantly lower than that of the control group and showed a trend of first increasing and then decreasing (Figure 3B).

### 2.4. Aphid Production in Three Successive Generations and the Expression of Vitellogenin (Vg) in Aphids

F1 aphid production was significantly lower in the 50 and 100 mg/L Zn groups than in the control. However, the production of F2 and F3 aphids in each Zn^2+^ treatment was not significantly different from that in the control (Figure 4A). F1 aphids’ relative expression level of *Vg* was significantly lower than that in the 50 mg/L Zn group. The relative expression of *Vg* was significantly higher in the 50 mg/L and 100 mg/L Zn group F2 aphids. No significant differences were observed in F3 aphids (Figure 4B).

### 2.5. Aphid Carbohydrates Content and Trehalase Activity

In the 1st generation of adult aphids, trehalose content initially decreased and then increased with increasing concentrations (Figure 5A). However, the overall content increased compared to that of the control group. The glucose content initially increased and then decreased (Figure 5B). Trehalase activity increased gradually, and there was a significant difference in the 150 mg/L Zn group (Figure 5D). Similarly, in the F2 aphids, the trehalose content in 50 and 100 mg/L Zn groups was significantly higher than that in the control and 150 mg/L Zn groups (Figure 5A), whereas the glucose content in 50 and 150 mg/L Zn was significantly lower than that in the control group, while there was no significant difference in the other groups (Figure 5B). In the 100 mg/L Zn group of the F3 aphids, the trehalose content of aphids was significantly lower than that of the control group, whereas the glucose content and soluble trehalase activity were significantly higher than those in the control group. Membrane-bound TRE activity was significantly lower in the 50 mg/L Zn group than in the control group (Figure 5B,E). The glycogen content of F1, F2, and F3 adult aphids was not significantly different between any of the groups (Figure 5C).

### 2.6. Expression of Trehalose Metabolism Related Gene in Aphids

In F2 adult aphids, the relative expression levels of *TRE* and *TPS* showed the same trend and were significantly higher in the 50 mg/L and 100 mg/L Zn groups than in the control group. However, in adult aphids from the 1st generation, the relative expression levels of *TRE* and *TPS* in the treatment groups were significantly lower than those in the control group except in the 150 mg/L Zn group. In adult aphids from the 3rd generation, the relative expression level of *TRE* was significantly lower in the Zn treatment group than in the control group, and the relative expression level of *TPS* was significantly lower in the 150 mg/L Zn group than in the control group (Figure 6).

### 2.7. Changes in the Content of Zn in Broad Bean Roots, Stems, and Leaves before and after Aphid Inoculation

We found that as the Zn concentration increased, the Zn content in the roots, stems, and leaves of broad beans also increased, indicating that Zn can accumulate in these parts. In the 150 mg/L Zn group, the Zn content in the roots of broad beans with aphid feeding was significantly higher than that without aphid feeding (Figure 7A). In the stems of broad beans, there was no significant difference between the groups before and after aphid feeding (Figure 7B). In broad bean leaves, except for the 50 mg/L Zn treatment group, the Zn content before aphid feeding was significantly lower than that in the non-feeding group (Figure 7C). These results indicate that the initial infection of aphids can alter the enrichment of heavy metals in broad beans, and the trend is the opposite in the roots and leaves.

## 3. Discussion

Many studies have shown that most heavy metals in the soil or water can be transferred to plants. Plant roots have a large absorption surface, which can absorb soluble heavy metals, while others are difficult to be absorbed [45]. Heavy metal stress can affect the growth of most plants to some extent [46]; incidentally, the tolerance of plants to heavy metal stress is a complex life process [47]. Generally, the metal absorption rate of leafy vegetables is higher than that of non-leafy vegetables [48]. The broad beans used in this study are categorized as leafy vegetables. The results showed that the Zn content in the broad bean roots reached 1000 mg/kg, which was far greater than the Zn content in the soil (Figure 1A). However, notably, Zn accumulation in the roots of broad beans treated with 150 mg/L Zn irrigation was limited (Figure 1B). It was speculated that when Zn concentration irrigation reaches a certain value, the accumulated Zn content in the roots may decrease. Metal transporters, such as OsZIP1 [49], promote the outflow of Zn, Cu, and Cd metal ions in rice, which may explain why the accumulation of Zn in rice did not increase; perhaps there are similar heavy metal transporters in fava beans. However, the concentrations used in this experiment were limited, and further in-depth research is required. The broad bean has a high tolerance to Zn, which is consistent with Liu’s results where Phyllostachys pubescens also showed no significant difference in seed germination rate under Zn stress compared to the control [50]. In tomato seedlings, the mass ratio of the single Zn stress increased, and the height of the tomato seedlings initially increased and then decreased [51]. Our results confirmed this finding (Table 1). This may be because the content of reactive oxygen species in plants increases in the low heavy-metal concentrations, which can activate proteases, regulate synthesis, induce gene expression, and promote cell division and proliferation [52]. High concentrations of heavy metals affect the absorption of nutrients by plants, causing them to dwarf and reduce biomass. Heavy metals destroy the integrity of the cell structure and affect the integrity and permeability of the cell plasma membrane [53].

Physiological changes in plants eventually result in morphological changes. During plant growth and development, chlorophyll is one of the main pigments used for photosynthesis [54], and a reduction in chlorophyll content inhibits the capture and utilization of light energy [6]. Zn is a component of various enzymes, such as oxidoreductases and transferase [55], and it plays a key role in the formation of carbohydrates, chlorophyll, and root growth [56]. However, excessive Zn is as toxic to plants as are other heavy metals. Toxicity in plants is mainly manifested by inhibiting the synthesis of chlorophyll, affecting photosynthesis, causing leaf chlorosis, accelerating the aging of plants, causing leaves to slowly turn yellow, and affecting the growth and development of plants [57]. The chlorophyll content of Dryopteris decreased significantly under heavy metal stress [47]. Vassilev also indicated that high Zn concentrations reduce the content of auxiliary photosynthetic pigments, including Chl a and Chl b, by interfering with the absorption and translocation of iron (Fe) and magnesium (Mg) in the chloroplasts of legumes under heavy metal stress [58]. However, in our experiment, except for the yellowing of the leaves, there was no other form of leaf damage, which is largely attributed to the homeostasis of the plant intracellular environment. Cells regulate their intracellular osmotic potential by accumulating soluble sugars and other osmotic substances and by increasing intracellular osmotic pressure to resist adverse environments [59]. Notably, in this study, the soluble sugar content in the roots of the Zn group was lower than that of the control group, which may be due to the fact that roots are more sensitive to stress compared to the aboveground parts, and they need to use sugar to obtain more energy to maintain basic life activities. However, in the leaves, the soluble sugar content increased with stress concentration, and there was no difference in the stems (Figure 3A). A previous study showed that the soluble sugar content of barley also increases with the accumulation of aluminum (Al), Cd, and copper (Cu) [60]. These results indicate that soluble sugar, as an osmoregulatory substance, helps to alleviate the toxic effects of high concentrations of Zn.

PRO is an important osmotic substance in plants that maintains intracellular homeostasis through osmotic regulation [61]. Notably, the role of PRO in heavy metal stress is multifaceted. Research has shown that plants grow in heavy metal environments, and functional groups, such as carboxyl, amino, thiol, and phenolic groups in free amino acids, can combine with metal ions to form stable chelates. For example, the in vitro protective effect of PRO against glucose-6-phosphate dehydrogenase is achieved through the formation of a non-toxic Zn-PRO complex [62], which can passivate and detoxify heavy metals. PRO, glutamic acid, and cysteine are also precursors of plant-chelating peptides in free amino acids and are considered to play an important role in plant heavy metal tolerance [63,64]. However, the formation of complexes between PRO and heavy metals is not the main function of PRO induced by heavy metals because the stability constant of metal–PRO complexes is too low to prevent the binding of heavy metals to enzymes [65]. PRO mainly reduces the toxicity of heavy metals by enhancing the reducibility of the cytoplasmic environment through its antioxidant properties rather than forming complexes with heavy metal ions. Sharma [62] believed that PRO induced by heavy metals plays a role mainly in osmotic regulation and avoids enzyme dehydration rather than chelating heavy metals. PRO accumulation has been observed under many biotic stresses but not in our experiment (Figure 3B). It is possible that different plants synthesize and degrade PRO differently in different environments; therefore, reports on the role of PRO are inconsistent and worthy of further research [61].

Heavy metals in soil are absorbed by plant roots and transported to different tissues through various transporters, thereby enhancing the absorption of heavy metals by plants [66,67]. Heavy metals enter the body of phytophagous insects mainly through the soil–plant pathway. For example, the growth, development, and reproductive ability of *Lymantria dispar* were found to be disturbed, which feeds on *Populus alba* × *P. berolinensis* and enriches heavy metals [68]. Moreover, heavy metals can weaken the reproductive abilities of insects. Accumulation of the heavy metal Cd weakens the oviposition ability of two kinds of *Tetranychus cinnbarinus* in tomatoes [69]; some studies have shown that heavy metal stress reduces the oviposition of *Spodoptera exigua* adults. Treatment with high concentrations of Zn reduced the oviposition, fecundity, and hatchability of adult females of *Spodoptera litura* [70]. In our study, there was no significant difference in the heavy metal stress of aphids produced by the F2 and F3 female aphids compared with the control group (Figure 4A), which may be due to the special absorption strategy of this species [71]. Another reason may be that the Zn in our study was mediated by plants, or it may be that certain substances produced by plants have a certain impact on direct feeding.

Trehalose, as the “blood sugar” of insects, plays an important role in various physiological activities. It has been reported that under dry conditions, the trehalose synthetase activity of *Drosophila melanogaster* increases, and trehalase activity decreases, resulting in an increase in trehalose content in the body [72]. In *Ostrinia nubilalis*, glycerol and trehalose are the most abundant cryoprotectants in diapause larvae [73]. Yu [17] found that under long-term Cd stress, the trehalose content of Aedes albopictus increased, whereas glucose content and trehalase activity decreased. In addition, under acute Cd stress, the trehalose metabolism of A. albopictus was affected [74]. When *Spodoptera litura* ingested an artificial diet supplemented with Cd, the trehalose content in adults decreased and the trehalase and trehalose synthetase activities were also affected [75]. In our study, the trehalose concentration in F1 and F2 increased at 50 and 100 mg/L Zn, whereas that in F3 decreased at 100 mg/L Zn (Figure 5). This may be due to the stress response when aphids were initially exposed to Zn stress, resulting in increased trehalose synthesis as a protective agent. When the aphids reached the 3rd generation, trehalose needed to be hydrolyzed into glucose to provide energy for aphids in order to maintain their own basal metabolism, as evidenced by changes in trehalose enzymes. In the 1st generation, the changes in glucose and trehalose content consistently increased because Zn may also directly regulate the transcription level of glucose regulatory proteins in aphids. Changes in sugar content alter the expression of genes involved in trehalose metabolism. *TRE* and *TPS* are the trehalase and trehalose synthase genes, respectively. In F1 aphids, owing to the accumulation of trehalose, *TRE* expression decreased. Similarly, the decrease in *TPS* expression may be due to feedback regulation by trehalose. In F2 aphids, the expression of *TRE* and *TPS* was increased to maintain stable trehalose levels (Figure 6).

Notably, the Zn content in the roots of broad beans with aphid feeding was significantly higher than that of non-aphid feeding in the 150 mg/L Zn group, while the Zn content in the leaves exhibited the exact opposite trend. Huang [66] suggested that insect feeding affected the accumulation of heavy metals in plants. In Arabidopsis thaliana, feeding *Pieris rapae* larvae also enhanced Cd accumulation in the leaves [76]. On the contrary, the cumulative level of Cd prevented the feeding of *Pieris rapae* [77]. Specifically, herbivorous animals induce defense responses related to plant physiological and morphological responses. To this end, scientists have proposed several hypotheses suggesting that this behavior may be beneficial. The most common is the elemental defense hypothesis, which suggests that heavy metals can serve as defenses for herbivores [78]. It is indicated that the feeding of aphids may also affect the accumulation of heavy metals in broad beans (Figure 7), but the specific mechanism needs to be clarified through a series of subsequent experiments. Therefore, broad bean plants can be a potential target for plant metal remediation because they can absorb heavy metal pollutants from the soil through their roots and accumulate them in the aboveground parts to reduce the concentration of heavy metals in the soil and achieve the goal of alleviating heavy metal pollution in the soil. Moreover, the initial feeding of aphids can promote enrichment of metals, providing a method for the subsequent repair of broad beans.

## 4. Materials and Methods

### 4.1. Insect Sources and Tested Plants

The broad beans (*V. faba*) type used in the experiment was Qingchan No. 14, and the aphid species was *M. crassicauda*. Broad bean seedlings were planted for feeding by *M. crassicauda*. The feeding conditions of the broad beans and aphids were: 19 ± 1 °C, humidity 70 ± 5%, photoperiod 14L:10D.

### 4.2. Establishment of Experiments

According to the research report [79], ZnSO_4_ was diluted with water to a Zn mass ratio of 0 (tap water only; control), 50, 100, and 150 mg/L. Broad bean seeds were soaked in Zn solutions for 24 h and then planted in soil (nutrient soil: vermiculite: perlite = 12:4:2). According to the growth requirements of broad beans, it is determined to water 400 mL corresponding concentration solution every 3 days. Based on the growth status of the broad bean, it is determined that on the 10th day after the broad bean is planted in the soil, normal aphids were transferred to the broad bean seedlings as F0 generation. Their offspring nymphs were considered the F1 generation. F1 aphid adults were then transferred to new broad bean seedlings (treated with the same Zn concentration), where they produced F2 nymphs; these were treated in the same way as F1 on reaching adulthood and persistent infecting to F3.

### 4.3. Measurement of Germination Rate and Development Observation of Broad Bean Seedling

The germination of broad beans was observed and recorded on days 7, 9, 11, and 13 after planting in the soil with 45 seeds per biological group. In addition, the height of broad bean seedlings was measured on days 9, 11, 13, and 15 after planting in the soil, and 10–30 broad beans were randomly selected from each biological replicate. The growth statuses of broad beans were observed and photographed with three biological replicates per group.

### 4.4. Determination of Zn Content in Broad Bean

Three broad bean growing sites near 30°17′50.84″ N and 119°59′42.9″ E, including campus, construction site, and vegetable field, were randomly selected. The soil, broad bean roots, and broad bean stems were collected from each sampling site. Inductively coupled plasma mass spectrometry (ICP-MS) was then used to determine the Zn content of the samples. Samples (0.2–0.5 g) were placed in a digestion tank with 4 mL nitric acid addition and pre-oxidized at 80 °C for 1 h. Then, they were digested using a microwave digestion instrument and transferred to a 50 mL volumetric flask with first-class water.

### 4.5. Determination of Soluble Total Sugar and Chlorophyll and PRO Content in Broad Beans

The roots, stems, and leaves of broad beans were collected on day 21 after planting. They were oven-baked at 110 °C for 15 min, which was then reduced to 70 °C overnight. Dried broad bean roots were ground and used to determine the total soluble sugar content. Leaves (the second pair from top to bottom) at the same position as fresh broad beans and irrigated with different Zn solutions on day 21 were used for chlorophyll content determination. The *Guide to Modern Plant Physiology Experiments* was used for specific operations and calculation methods. The PRO content was determined using the Nanjing Jiancheng Reagent Kit (Nanjing, China, proline assay kit). Please refer to the manufacturer’s instructions for the methods of determination.

### 4.6. Aphid Fecundity Determination

When the broad bean seedlings grew to approximately 4–5 cm, normal adult aphids moved toward the broad bean seedlings. Ten days after infection, F1 adult aphids were collected and transferred to broad bean seedlings and treated with a fresh Zn solution at the corresponding concentration (50 mg/L, 100 mg/L, 150 mg/L). Aphid production was measured every 24 h for 7 d. Aphid production in the three batches of female aphids was counted.

### 4.7. Determination of Carbohydrate Content and Trehalase Activity in Aphids

This part of the experiment was conducted according to the method described by Zhang [80].

### 4.8. Total RNA Extraction and cDNA First Strand Synthesis of Aphids

Total RNA was extracted from *M. crassicauda* using the RNAiso Plus kit (Invitrogen, Carlsbad, CA, USA) according to the manufacturer’s instructions. Subsequently, RNA integrity was detected using 1% agarose gel, and the concentration and purity of the extracted RNA were determined using a NanoDrop™ 2000 (Waltham, MA, USA). A PrimeScript™ RT Reagent Kit with gDNA Eraser (Takara, Kyoto, Japan) was used for reverse transcription of the cDNA.

### 4.9. Real-Time Fluorescence Quantitative PCR (qRT-PCR)

The qRT-PCR reaction system (TaKaRa TB Green^®^ Premix Ex TaqTM) was used as follows: 1 μL of cDNA, 5 μL of TB Green, 0.4 μL of forward primer, 0.4 μL of reverse primer, and 3.2 μL of ddH_2_O. The qRT-PCR reaction procedure was: pre-denaturation at 95 °C for 30 s, 40 cycles of 95 °C for 5 s, 55–60 °C for 30 s. *M. crassicauda* actin-mRNA was an internal control, and the primer sequences are shown in (Table 2) (the partial sequences we cloned of the three genes are provided in the Appendix A). The qRT-PCR data were analyzed using the 2^−ΔΔCT^ method [81].

### 4.10. Statistical Analysis

Insects were randomly divided into three replicates for each treatment. Results were expressed as the mean ± standard deviation (SD) or the mean ± standard error (SE) of independent replicates (n ≥ 3). The data were analyzed using IBM SPSS statistics 26 software. Statistical significance was defined as *p* < 0.05. Tukey’s test of one-way ANOVA was performed to test the significance of differences among treatments. All figures and tables were produced using Office 2021 and GraphPad 8.0 software.

## 5. Conclusions

In summary, when we simulated soil Zn pollution, broad beans have a higher ability to accumulate Zn. Under Zn stress, broad beans maintained internal environment stability by regulating the content of soluble sugars and proline, which are permeable substances in their bodies. The height of broad bean seedlings had a phenomenon of low promotion and high inhibition, and the chlorophyll content in leaves decreased. However, the germination rate of broad beans was not significantly affected. In addition, Zn not only negatively affects fertility by reducing the egg production of the aphids but also has an impact on the trehalose metabolism. Finally, it is worth noting that the initial infection of aphids can alter the Zn enrichment ability of broad bean tissue. Our research results provide a theoretical basis for exploring the impact of soil heavy metal pollution on leguminous plants and their herbivorous aphids in the future, but further experiments are needed to explore the reasons for this in order to reveal a clearer response mechanism of heavy metal reactions.

## Figures and Tables

**Figure 1 ijms-24-09659-f001:**
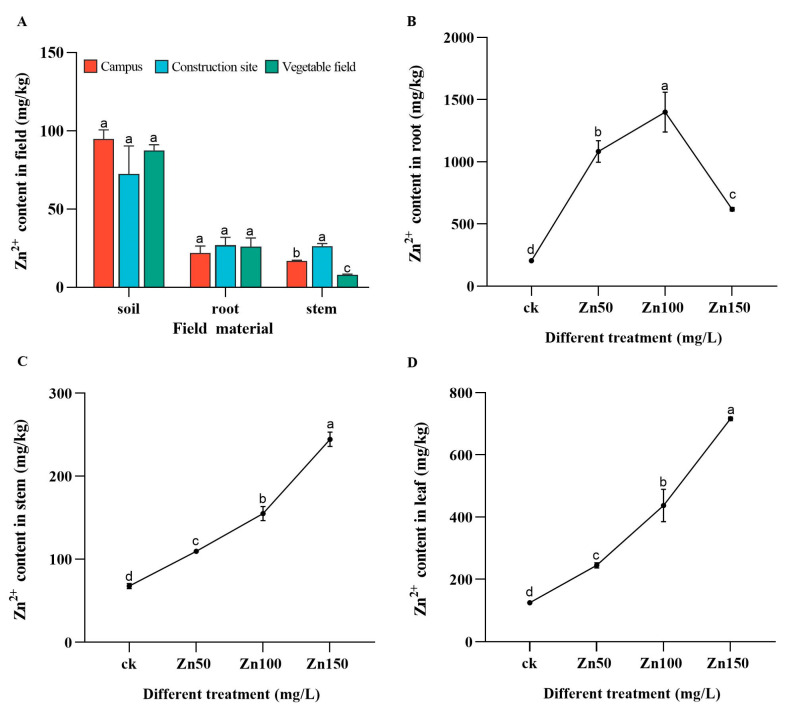
Zn content in soil and broad bean (root and stem) at three random sites of Campus, Construction site, and Vegetable field (**A**). Zn content accumulation in broad bean root (**B**), stem (**C**), and leaf (**D**) after irrigation with different concentrations of Zn^2+^ solution after 21 days. Bars represent means (+SD) of three replicate experiments. Bars with different letters indicate significant differences (one-way ANOVA, *p* ≤ 0.05).

**Figure 2 ijms-24-09659-f002:**
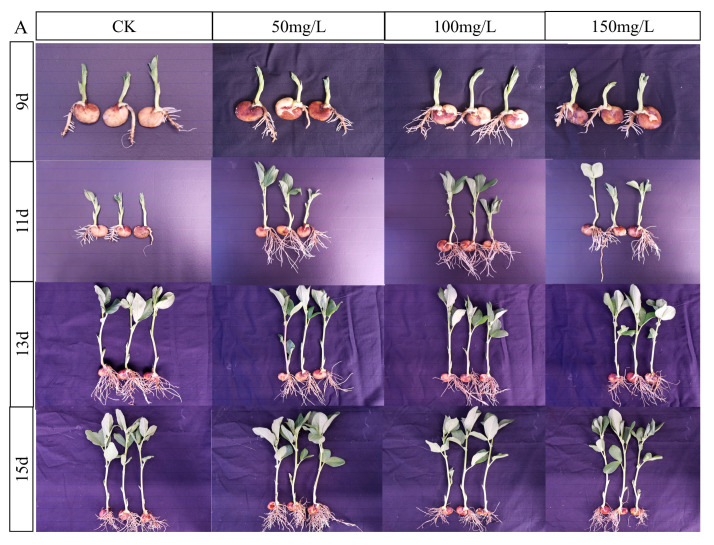
Changes in growth (**A**) and germination rate (**B**) after 7, 9, 11, and 13 days, and chlorophyll content (**C**) after 21 days of broad bean planting under Zn stress. Bars represent means (+SD) of three replicate experiments. Tap water was used as the control group. Asterisk indicates significant difference between Zn stress and control groups (*t*-test, * *p* ≤ 0.05).

**Figure 3 ijms-24-09659-f003:**
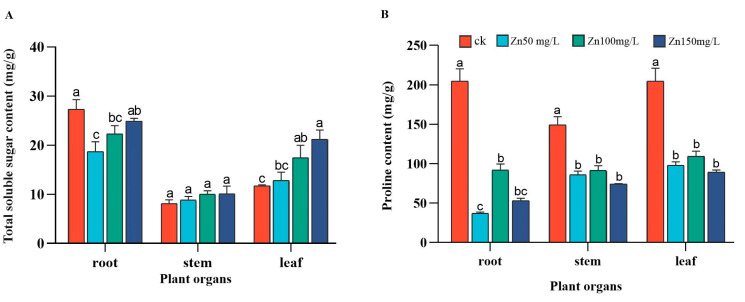
Changes in the content of soluble total sugar and PRO in broad bean roots, stems, and leaves after 21 days of Zn stress. Total soluble sugar content of broad bean tissue in different groups (**A**). PRO content of broad bean tissue in different groups (**B**). Bars represent means (+SD) of three replicate experiments. Tap water was used as the control group. Bars with different letters indicate significant differences (one-way ANOVA, *p* ≤ 0.05).

**Figure 4 ijms-24-09659-f004:**
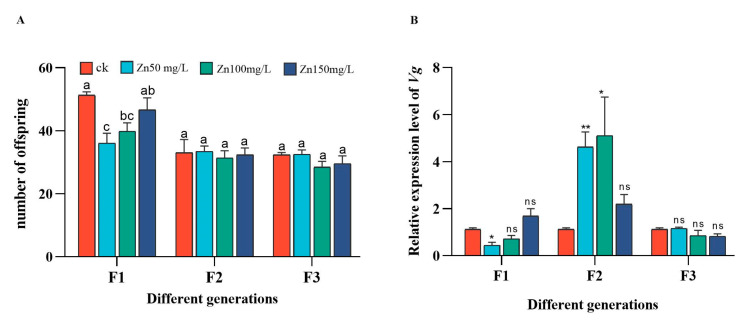
Postpartum algebra of single female aphids in different groups after Zn stress (**A**). The egg production of aphids in three generations changed (F1, F2, and F3 represent three generations). Changes of mRNA expression levels of *Vg* (**B**) in different generations of aphids, McrActin expression level and RT-qPCR were used to calculate the relative expression level of target gene. The bar graph shows the average value of three replicates (+SD). The control group was irrigated with tap water. Bar charts with different letters indicate significant differences (one-way ANOVA, *p* ≤ 0.05; the gene expression level of aphids grown under tap water was used as the control group. Asterisk indicates the significant difference between the Zn stress and control groups (*t*-test, ** *p* ≤ 0.01; * *p* ≤ 0.05).

**Figure 5 ijms-24-09659-f005:**
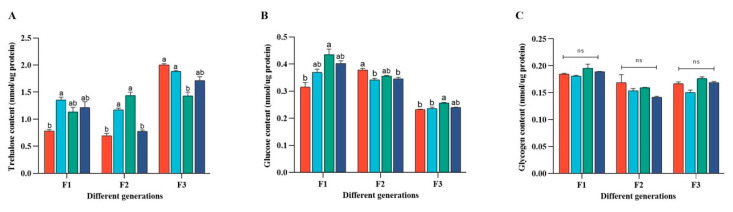
Changes of carbohydrate content in different generations of aphids after 21 days of Zn stress. (**A**) represents the change of trehalose content during stress, (**B**) represents the change of glucose content during Zn stress, (**C**) represents the change of glycogen content during Zn stress, (**D**) represents the change of soluble trehalase activity during Zn stress, and (**E**) represents the change of membrane-bound trehalase activity during Zn stress. Bars represent means (+SD) of three replicate experiments. Tap water was used in the control group. Bars with different letters indicate significant differences (one-way ANOVA, *p* ≤ 0.05).

**Figure 6 ijms-24-09659-f006:**
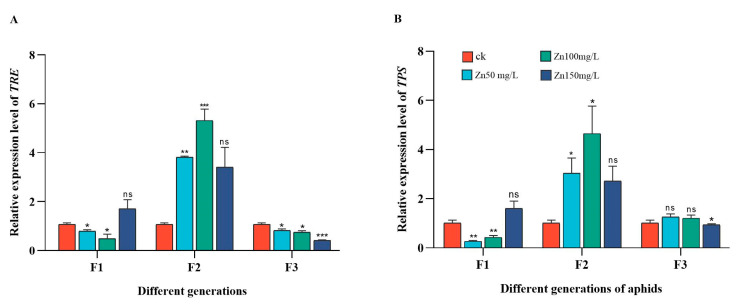
Changes of mRNA expression levels of *TRE* (**A**), *TPS* (**B**) in different generations of aphids (F1, F2, and F3 represent three generations), McrActin expression level and RT-qPCR were used to calculate the relative expression level of target genes. The bar graph represents the average of three replicates (+SD). The gene expression level of aphids grown under tap water was used as the control group. Asterisk indicates the significant difference between the Zn stress and the control groups (*t*-test, *** *p* ≤ 0.001; ** *p* ≤ 0.01; * *p* ≤ 0.05).

**Figure 7 ijms-24-09659-f007:**
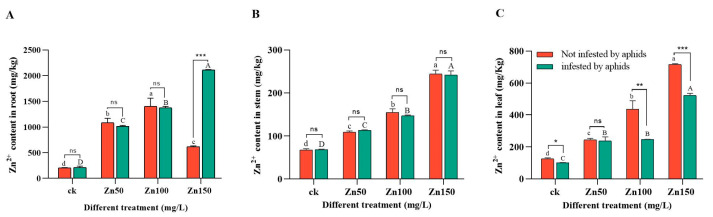
Changes in the content of Zn in broad bean roots, stems, and leaves after 21 days of Zn stress, (**A**) represents the change in the content of Zn enrichment in broad bean roots during the Zn stress period, (**B**) represents the change in the content of Zn enrichment in broad bean stems during the Zn stress period, and (**C**) represents the change in the content of Zn enrichment in broad bean leaves during the Zn stress period. Bars represent means (+SD) of three replicate experiments. Bars with different letters (Capital letters indicate differences between different treatment of the same tissue after aphid inoculation, while lowercase letters indicate differences between different treatment without aphid inoculation) indicate significant differences (one-way ANOVA, *p* ≤ 0.05). Tap water was used as the control group. Asterisk indicates a significant difference between infection and no infection by aphids (*t*-test, *** *p* ≤ 0.001; ** *p* ≤ 0.01; * *p* ≤ 0.05).

**Table 1 ijms-24-09659-t001:** Height of different groups of broad beans.

Days after Planting (d)	0 mg/L Zn	50 mg/L Zn	100 mg/L Zn	150 mg/L Zn
9	3.66 + 0.35 c	4.73 + 0.35 a	3.98 + 0.56 bc	4.26 + 0.37 b
11	8.06 + 0.46 b	8.84 + 0.49 a	8.14 + 0.41 b	7.48 + 0.36 c
13	12.95 + 0.55 b	14.14 + 0.91 a	11.57 + 1.22 c	12.22 + 0.53 bc
15	17.21 + 0.91 b	18.65 + 0.84 a	15.41 + 1.58 c	16.82 + 0.55 b

(Note: The difference between different letters in Table 1 is the comparison between different heavy metal treatments at the same time, the unit is cm; each point is the mean ± SD of three replicates).

**Table 2 ijms-24-09659-t002:** Primers used in real-time polymerase chain reaction.

Gene Name	Primer Name	Nucleotide Sequences (5′-3′)
*TPS*	McTPS-F	CGTGGACAGGCTAGACTACA
	McTPS-R	CAGCTCAGTCTCGTCCTTGA
*TRE*	McTRE-F	TGGCAAGATACTACGCACCA
	McTPS-R	ATCAGCCAATACCCCACGAT
*Vg*	McVg-F	GCATTAGCCACTATGTTTCA
	McVg-R	CGTATTGCTCCATTGTTGT
Actin	McActin-F	GATCATTGCCCCACCAGAAC
	McActin-R	TTTACGGTGGACAATGCCTG

## Data Availability

The data presented in this study are available upon request from the corresponding author.

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
