# Peer review of "Regulation of *Vicia faba* L. Response and Its Effect on *Megoura crassicauda* Reproduction under Zinc Stress"

_ijms, 2023, doi:10.3390/ijms24119659_

Round 1

Reviewer 1 Report

-         Shorten the abstract, concise the explanation of the experiment and outcomes here. Present the organization of the manuscript before section II.

-          List the significance & contributions of this study and key improvement over the existing research works at the end of section I.

-          Introduce a new subsection named experimental setup under section II.

-          What are the constraints and complexity of measuring the effects of excessive Zn stress on broad beans and trehalose metabolism of aphids.

-          Summarize the paper in the conclusion section. 

Author Response

Dear reviewer:

Thank you again for your hard work in reviewing our manuscript and your suggestions. All your suggestions are very important, and they are of great guiding significance for my paper writing and scientific research! Now we have carefully revised the manuscript according to your suggestions, and all revised chapters are marked in red. I will upload the modified specific information in the form of attachment.

Best regards!

Reviewer 2 Report

The article was written correctly. The aim was to evaluate the stress of broad bean plants caused by different doses of Zn in the amounts of 50, 100 and 150 mg/kg. Roots, leaves and stems were evaluated. The population of aphids was also analyzed depending on the experimental factors. The experiment was performed correctly. Shown in charts and figures. Appropriate statistical methods were used to compile the results. The inference and discussion are correct.

However, the work contains some errors.

1. None and it is necessary to add a research hypothesis.

2. Broad bean is a spring plant, not a winter plant (line 70), and should be corrected. Justify the possibility of using Vicia Faba L in phytoremediation.

3. Please correct and use SI units in your work (not mg/kg)

4. Describe the layout of the experiment and the statistical methods used for the calculations.

5. Be sure to add a conclusion chapter. Summarize the results of your research in a few sentences.

Interesting article, but needs more improvement. After correcting errors, it can be published in Int. J. Mol. Sci.

English is understandable

Author Response

(The authors gave the same response as above.)

Round 2

Reviewer 1 Report

Authors have revised the manuscript thoroughly.